# Effects of Starch Overload and Cecal Buffering on Fecal Microbiota of Horses

**DOI:** 10.3390/ani12233435

**Published:** 2022-12-06

**Authors:** Caio C. Bustamante, Vanessa B. de Paula, Isabela P. Rabelo, Camila C. Fernandes, Luciano T. Kishi, Paulo A. Canola, Eliana Gertrudes de M. Lemos, Carlos Augusto A. Valadão

**Affiliations:** 1Department of Veterinary Medicine and Surgery, School of Agricultural and Veterinary Sciences, UNESP-São Paulo State University, Jaboticabal 14884-900, SP, Brazil; 2Department of Technology, Multiuse Sequencing Laboratory, School of Agricultural and Veterinary Sciences, UNESP-São Paulo State University, Jaboticabal 14884-900, SP, Brazil; 3Department of Technology, Biochemistry of Microorganisms and Plants Laboratory, School of Agricultural and Veterinary Sciences, UNESP-São Paulo State University, Jaboticabal 14884-900, SP, Brazil

**Keywords:** dysbiosis, equine, feces, NSG, Ruminococcaceae, starch overload, *Streptococcus*

## Abstract

**Simple Summary:**

**The purpose of this study was** to determine if the intracecal injection of alkaline solution (buffer; Mg(OH)_2_ + Al(OH)_3_) could stabilize fecal microbiota and clinical changes in horses submitted to starch overload. Clinical signs, gross analysis of the feces, and fecal microbiota were evaluated for 72 h (T0; T8; T12; T24; T48; T72) from ten crossbred horses. Horses were allocated to group I (water–saline and starch–buffer treatments) and group II (water–buffer and starch–saline treatments). Starch overload reduced the richness and diversity of the fecal microbiota. However, the starch–buffer treatment led to a greater increase in amylolytic bacteria and decrease in fibrolytic bacteria than did the starch–saline treatment. This study showed that cecal infusion of buffer did not prevent the intestinal disturbances and their consequences.

**Abstract:**

Starch overload in horses causes gastrointestinal and metabolic disorders that are associated with microbiota changes. Therefore, we identified the fecal microbiota and hypothesized that intracecal injection of alkaline solution (buffer; Mg(OH)_2_ + Al(OH)_3_) could stabilize these microbiota and clinical changes in horses submitted to corn starch overload. Ten crossbred horses (females and geldings) were allocated to group I (water–saline and starch–buffer treatments) and group II (water–buffer and starch–saline treatments). Clinical signs, gross analysis of the feces, and fecal microbiota were evaluated through 72 h (T0; T8; T12; T24; T48; T72). Corn starch or water were administrated by nasogastric tube at T0, and the buffer injected into the cecum at T8 in starch–buffer and water–buffer treatments. Starch overload reduced the richness (*p* < 0.001) and diversity (*p* = 0.001) of the fecal microbiota. However, the starch–buffer treatment showed greater increase in amylolytic bacteria (*Bifidobacterium* 0.0% to 5.6%; *Lactobacillus* 0.1% to 7.4%; *p* < 0.05) and decrease in fibrolytic bacteria (Lachnospiraceae 10.2% to 5.0%; Ruminococcaceae 11.7% to 4.2%; *p* < 0.05) than starch–saline treatment. Additionally, animals that received starch–buffer treatment showed more signs of abdominal discomfort and lameness associated with dysbiosis (amylolytic *r* > 0.5; fribolytic *r* < 0.1; *p* < 0.05), showing that cecal infusion of buffer did not prevent, but intensified intestinal disturbances and the risk of laminitis.

## 1. Introduction

The large intestine of horses is an anaerobic fermentation chamber filled with diverse microorganisms [1] that play an essential role in protection against pathogen overgrowth, stimulation of the immune response in the gut, and enhanced intestinal barrier function [2,3,4]. This microbiota has been shown to be influenced by various factors, such as age, management, breed, and environment [5,6]. Similarly, differences in the fecal bacterial communities between horses with gastrointestinal diseases and their healthy counterparts have been reported [5,7,8]. Therefore, a disturbance in the equine hindgut microbiota can alter the fermentation patterns and ultimately lead to metabolic disorders, such as laminitis [2,9,10,11].

Diet can negatively modulate the intestinal microbiome [1,12] since sudden dietary changes and high-starch diets are common practices in intensive horse breeding [13]. A previous study [14] revealed that the abrupt inclusion of high cereal content in the equine diet caused dysbiosis. It appears that a substantial amount of cereal grain escapes enzymatic digestion in the foregut [15]; consequently, excessive fermentation occurs in the hindgut, leading to pH changes. The pH changes in the digestive system have already been reported to cause microbiota disruption [7,16], also shown in horses’ intestinal microbiota submitted to starch overload [9,10,17].

A low pH in the gut lumen alters the microbiota due to lactate accumulation, which can induce subclinical acidosis [18]. The death of acid-sensitive bacteria is associated with the release of endotoxins that can compromise the intestinal barrier function, developing potentially life-threatening complications in horses [19,20,21]. Strategies to stabilize the fermentative process, avoiding great intestinal pH variations can minimize such complications arising from poor food management. Compounds containing magnesium and aluminum hydroxide (Mg(OH)_2_ + Al(OH)_3_) have been reported to have therapeutic value in humans and milk-fed calves as antiacid substances to increase the gastrointestinal pH [16,22,23]. In horses, this antiacid is helpful for treating gastritis [24] and large bowel impaction [25]. As an alkaline and cathartic product, Mg(OH)_2_ + Al(OH)_3_ increases pH and water influx into the lumen, pushing the impacted compound through the hindgut until it is eliminated [26].

Most studies related to the Mg(OH)_2_ + Al(OH)_3_ solution in animals have been conducted to evaluate its effects in gastrointestinal pH alterations [23,24,27]. Moreover, these studies administered the alkalizing agent orally. The effect of Mg(OH)_2_ + Al(OH)_3_ on intestinal dysbiosis following starch overload in horses has not been evaluated yet. Therefore, we identified the fecal microbiota and hypothesized that it, and consequently, clinical changes induced by the corn starch overload, could be stabilized by the intracecal injection of this alkaline solution.

## 2. Material and Methods

### 2.1. Animals, Facilities, and Management

This study was approved and supervised by the committee of animal use and care of the São Paulo State University (23391/15). We used 10 crossbred horses (geldings and mares), aged 13 ± 5.6 years and weighing 353 ± 28 kg. Horses were considered healthy based on physical examination and hematology. All horses were submitted to typhlopexy in the right flank [28] at least 90 days prior to the study. After 21 days post-typhlopexy, cecal catheterization was performed under sedation using xylazine (0.5 mg/kg IV) and local anesthesia with lidocaine in order to fix a Levin tube number 18 into the cecum’s lumen. During this period, the horses were housed in separate stalls with partial access to a paddock; and were fed with 2 kg/day of commercial ration (Selvagem^®^—Agromix—Jaboticabal, São Paulo, Brazil), 4 kg/day of *coast cross* (*Cynodon dactylon*) hay, and free access to fresh water and mineral salt. Seven days after the catheterization, the animals were restricted to the stalls, and the volume of hay was increased to 8 kg/day, with a period of adaptation of two weeks to start the study.

### 2.2. Experimental Design

The animals were randomly distributed into two groups (I and II) of five horses. Each group was submitted to two different treatments (group I: water–saline [WS], and starch–buffer [SB]; group II: water–buffer [WB], and starch–saline [SS]; Table 1). To this end, a washout period of 15 days (minimum) was allowed between the treatments from the same group. The experimental evaluations were performed in time measurements from the baseline (T0), and T8, T12, T24, T48, and T72 h after T0.

At the baseline, water or corn starch overload were administered by nasogastric tube, and subsequently at T8, buffer and/or saline 0,9% were administered intracecally.

### 2.3. Procedures

Physical examination (heart rate, respiratory rate, rectal temperature), evaluation of signs of lameness and abdominal pain, water and food intake, and fecal production were evaluated at T0 to T72. In addition, hydration status was estimated by evaluating the oral mucosa characteristics and skin tenting. Feces production was subjectively evaluated for consistency as firm (normal), pasty (cow pie consistency), or liquid (diarrhea) [30]. We also assessed the feces’ odor (characteristic or fetid) and color (normal or white). Evidence of laminitis was assessed using Obel grading [31] by encouraging animals to walk on a six-meter flat cement floor, observing reluctance to walk, and signs of lameness. Horses with Obel grade 2 or more were immediately treated with anti-inflammatory and analgesic drugs until no more signs of pain were evident.

Following clinical evaluation, fecal samples were collected via rectal ampulla with gloved hands at each time point, placed into sterile tubes and stored at −18 °C until processing. Bacterial DNA was extracted as previously described [32].

### 2.4. Data Processing

Microbial DNA was extracted using the PowerLyzer^®^ PowerSoil^®^ DNA Isolation Kit (MO BIO Laboratories Inc., Solana Beach, CA, USA), according to the manufacturer’s protocols. DNA integrity was checked in 1% agarose gels, with 5 µL of metagenomic DNA and 3 µL of carrying buffer (0.025% of bromophenol blue [p:v], and 50% of glycerol [*v*:*v*]) submitted to horizontal electrophoresis with 1X TBE buffer (Tris-89 mM, boric acid-89 mM, EDTA-2.5 mM, pH 8.3), and stained by sybr gold (0.5 µg/mL; Thermo Fisher Scientific, MA, USA for approximately 2 h at 80 V. Metagenomic DNA quality was evaluated in a NanoDrop ND-1000 spectrophotometer (Thermo Fisher Scientific, Waltham, MA, USA) at 260 nm and 280 nm, with the 260/280 ratio being used to estimate DNA purity. DNA concentration was determined using a Qubit^®^ 2.0 fluorometer (Thermo Fisher Scientific, MA, USA) and a Qubit™ DNA BR Assay Kit (Thermo Fisher Scientific, Waltham, MA, USA), following the manufacturer’s recommendations.

After extraction, the V4–V5 regions of the bacterial 16S rRNA were amplified using polymerase chain reaction (PCR; 95 °C for 3 min, followed by 35 cycles at 95 °C for 30 s, 55 °C for 90 s, 72 °C for 45 s, and a final extension at 72 °C for 5 min). The primers 515F (5′-GTGCCAGCMGCCGGCCTAA-3′) and 926R (5′-GGACTACHVGGGTWTCTAAT-3′) [33] were used for that purpose. PCR reactions and integrity of the amplified fragments were performed according to Fernandes et al. [34]. Then, the PCR products were purified using the Zymoclean™ Gel DNA Recovery Kit w/Zymo-Spin™ IC columns (Zymo Research, Irvine, CA, USA), following the manufacturer’s instructions. Quantification of the samples/library was conducted with the KAPA Library Quantification Kit Illumina^®^ (Integrated DNA Technologies, NC, USA) and quantitative PCR, as instructed by the manufacturer. Purified and quantified libraries were pooled in equimolar quantities and sequenced (2 × 250 bp) using the MiSeq^®^ reagent kit v2 (500 cycles), according to the manufacturer’s protocols using the Illumina platform.

### 2.5. Bioinformatics and Statistical Analyses

Data were submitted to bioinformatic analysis using the Quantitative Insights into Microbial Ecology (QIIME) package, version 1.9.1 [33]. Sequences smaller than 200 bp and those detected with chimeric sequences were eliminated. The Ribosomal Database Project II [35] was used to align high-quality sequences, obtain the taxonomic designation and the relative abundance of different bacterial groups. The resulting alignments were used to calculate the distance matrix and group the sequences in operational taxonomic units (OTUs) provided by QIIME. Kruskal–Wallis statistical analysis was employed to evaluate the differences amongst richness and alpha diversity (Shannon index) of samples in each treatment using the MicrobiomeAnalyst tool [36]. A weighted version of UniFrac, coupled with principal coordinate analysis (PCoA), was conducted to assess differences in the community structure (beta diversity) amongst treatments and data collection periods. Interactions between treatments and times were assessed with analysis of variance (ANOVA) followed by Tukey’s post-hoc test. The PROC MIXED tool from SAS software (SAS Inst. Inc., Cary, NC, USA) was used for that purpose. The Pearson’s correlation coefficient was applied to the changes in fecal microbiota and the animal’s clinical evaluation using GraphPad Prism Software version 7.00 (La Jolla, CA, USA). The level of significance was set at *p* < 0.05 for all analyses.

## 3. Results

### 3.1. Clinical Findings

Animals that received WS (water–saline) and WB (water–buffer) treatments did not show behavioral and systemic changes. On the other hand, those that received SB (starch–buffer) and SS (starch–saline) treatments showed increased heart rate starting at T8 and T12 up to T72, respectively. In addition, rectal temperature spiked after starch overload, between T8 and T48 (Appendix A).

Signs of gastrointestinal discomfort, characterized by diarrhea, were observed in all animals that received SB treatment. Also, 12 to 48 h following starch overload, fecal aspect drastically changed from normal consistency, smell, and color to a soft (pasty) consistency, fetid smell, and white color (Appendix A). Horses in the SS treatment group developed diarrhea in the first 24 h. Dehydration occurred from T12 to T48 and loss of appetite from T8 to T12 only in animals in the SB treatment group (Appendix A).

Forty-eight hours after starch overload, two animals in the SS treatment group presented Obel grades 2 and 3 lameness, characterized by reluctance to move, as the number of steps increased (from 8 to 10 steps). Horses in the SB treatment group increased the number of steps (from 10 to 14 steps) from T24 onwards. One animal refused to move, while the other three showed slowness and discomfort during the steps, characteristic of Obel grades 2, 3, and 4 (Appendix A).

### 3.2. Sequencing Information and Comparison of the Bacterial Communities

Of the 120 samples sequenced, only 118 were analyzed because two of them showed low sequencing quality. A total of 1,493,054 sequences were analyzed, using 12,653 sequences/sample to normalization, which identified 4,100 OTUs with 97% sequence similarity. These were assigned to 18 bacterial phyla, 108 families, and 190 genera. From these genera, 79.47% were classified as genus, and 20.53% were identified according to taxonomic hierarchy. Lower diversity (*p* = 0.001) and richness (*p* < 0.001) of fecal microbiota were observed in animals exposed to starch. These changes occurred from T12–T72, and between T24 and T72 in animals from the SB and SS treatment groups, respectively (Figure 1). The bacterial communities’ similarity (β-diversity) was associated with treatments and sampling intervals (Figure 2). The starch treatments (SB and SS) showed different β-diversity from other treatments (WS and WB), especially after 12 h.

### 3.3. Taxonomic Composition of Bacterial Communities

The bacterial phyla, family, and genera levels were characterized for each treatment, according to the sampling time. Only those with a total relative abundance greater than 1% were evaluated (Figure 3). The microbiota of SS- and SB-treated animals differed from the microbiota of WS- and WB-treated animals starting at T12, with some specific variations until T72 (Appendix A).

Firmicutes (42.0%), Bacteroidetes (3.7%), Euryarchaeota (3.2%), Actinobacteria (2.6%), Spirochaetes (2.2%), and Proteobacteria (1.1%) phyla were prevalent for all treatments. The SB treatment data presented increased relative abundance of Bacteroides (*p* < 0.001) and Actinobacteria (*p* = 0.011), starting at T48; and the Firmicutes (*p* < 0.001) phylum increased between T12 and T24. However, the abundance of Spirochaetes (*p* < 0.001) and Euryarcheota (*p* = 0.006) phyla was decreased at T12–T72 and T48–T72, respectively. The samples from SS showed prevalence of Proteobacteria (*p* = 0.05) between 24 and 48 h, and Firmicutes (*p* < 0.001) at 48 h, after starch overload (Appendix A).

The family with the greatest relative abundance for all treatments was Streptococcaceae (12.3%), specifically, the *Streptococcus* genus (12.2%). The most prevalent subcluster of the Lachnospiraceae family (9.1%) was *Clostridium XIVa* (4.9%), followed by a non-identified genus (2.1%). Another non-identified and more prevalent genus (4.5%) of the Ruminococcaceae family (9.7%) was also detected. *Clostridium III* (1.8%) and *Ruminococcus* (1.3%) comprised the other two more abundant genera. Just one genus predominated in the families Lactobacillaceae (*Lactobacillus*—3.8%), Prevotellaceae (*Prevotella*—2.9%), Methanobacteriacae (*Methanobrevibacter*—2.9%), Spirochaetaceae (*Treponema*—2.2%), Eubacteriaceae (*Eubacterium*—1.2%), Bifidobacteriaceae (*Bifidobacterium*—1.9%), and Erysipelotrichaceae (non-identified genus—1.6%). The families’ relative abundance variations were also observed following starch overload and/or buffer administration relative to their respective genera (Appendix A).

After 48 hours of SB treatment, the microbiota showed an increase in the abundance of *Bifidobacterium* (*p* = 0.021), *Prevotella* (*p* = 0.003) and a non-identified genus from the Erysipelotrichaceae (*p* < 0.001) family. The *Lactobacillus* (*p* = 0.001) genus increased only at T24, while *Streptococcus* (*p* = 0.001) increased at T12–T24. On the other hand, *Clostridium XVIa* (*p* < 0.001), *Eubacterium* (*p* < 0.001), *Ruminococcus* (*p* = 0.05), *Clostridium III* (*p* < 0.001), and a non-identified genus of the Ruminococcaceae (*p* = 0.001) and Lachnospiraciaea (*p* = 0.002) families showed decreased relative abundance between T12 and T72, which was also observed for *Treponema* (*p* < 0.001) and *Methanobrevibacter* (*p* = 0.003) genera at T24–T72 and T48–T72, respectively. Samples from SS animals showed an increase in the *Lactobacillus* (*p* = 0.001) genus at T24, and a decrease in abundance of *Ruminococcus* (*p* = 0.05; T24–T72), *Clostridium III* (*p* < 0.001), and *Eubacterium* (*p* < 0.001) genera (T24–T48; Appendix A).

### 3.4. Microbial Composition x Clinical Alterations

Data obtained from SS treatment showed a positive and moderate correlation between the prevalence of Firmicutes (*r* = 0.5; Pearson) and Proteobacteria (*r* = 0.6; Pearson) phyla and the clinical signs of diarrhea, inappetence, and dehydration (Appendix A). In addition, a moderate (*r* > 0.5; Pearson) and positive correlation between the prevalence of the Lactobacillaceae family, its genus (*Lactobacillus*), and clinical signs of diarrhea and inappetence was revealed. The Eubacteriaceae family and its genus (*Eubacterium*) correlated negatively with diarrhea and dehydration (*r* < 0.1; Pearson), as well as did the *Clostridium III* genera. This one also had a negative (*r* = −0.4; Pearson) correlation with number of steps (Appendix A).

Data from SB revealed a positive (*r* = 0.7; Pearson) correlation between the Obel grade (2 to 4) and an increase in the abundance of the Actinobacteria phylum. This abundance also had a moderate and positive (*r* > 0.4; Pearson) correlation with heart rate, decubitus and dehydration. There was also a positive (*r* > 0.5; Pearson) correlation between diarrhea, dehydration, and the increase in Firmicutes abundance. A negative (*r* = −0.4; Pearson) correlation occurred between the prevalence of Bacteroidetes phylum and dehydration; and between the decrease of Spirochaetes phylum and heart rate (Appendix A). The Ruminococcaceae and Eubacteriaceae families also correlated negatively with diarrhea (*r* = −0.7; Pearson). For these and the Lachnospiraceae family, as well as their respective genera (*Ruminococcus*, *Clostridium_III*, *Clostridium_XIVa* and *Eubacterium*), a negative correlation was observed between their prevalence and most of the clinical alterations. Moreover, the decrease in the Spirochaetaceae family and its genus (*Treponema*) had a negative correlation (*r* = −0.4; Pearson) with heart rate. On the other hand, *Lactobacillus* and *Streptococcus*, the most representative genera of the Lactobacillaceae and Streptococcaceae families, respectively, correlated moderately and positively (*r* > 0.5; Pearson) with diarrhea. In addition, the Bifidobacteriaceae family, its genus (*Bifidobacterium*), and a non-identified genus of the Erysipelotrichaceae family, in addition to a moderate correlation with heart rate (*r* > 0.4; Pearson), showed a strong (*r* > 0.7; Pearson) and positive correlation with the increase in Obel grade and the number of steps (Appendix A).

## 4. Discussion

Horses exposed to corn starch overload showed reduction in the diversity and richness of the fecal microbiota, characterized by proliferation of amylolytic bacteria, to the detriment of fibrolytic bacteria. It is well known that abrupt dietary changes involving starch-based concentrates affect normal fermentation in horses’ hindgut [37], cause colic [3,38] and laminitis [1,39,40], and reduce the microbiota’s diversity [41,42], increasing proliferation of amylolytic bacteria [43]. However, there were considerable differences in microbiota changes between starch–saline (SS) and starch–buffer (SB) treatments, since the latter showed more dysbiosis, even in the presence of the cecal buffer. Indeed, it was observed in our study that only the *Lactobacillus* genus increased in the SS treatment, while more than one genus of amylolytic bacteria prevailed in the SB treatment, such as *Streptococcus* and *Bifidobacterium*. The same was observed for fibrolytic bacteria, of which only three genera decreased in abundance in the SS treatment, and five genera decreased in the SB treatment. Nevertheless, all of them were from Lachnospiraceae, Rumicoccaceae, and Eubacteriaceae families.

In spite of *Lactobacilli* and *Bifidobacteria* generally being considered probiotic and health-promoting [21,44], some studies have reported higher prevalence of these bacteria in patients with gastrointestinal tract disease [45,46]. In our study we noticed that animals administered starch overload treatment presented clinical changes, such as diarrhea and inappetence, which were positively correlated with the prevalence of these taxa. It is well known that starch overload increases production of lactic acid by amylolytic bacteria, mainly *Lactobacilli*, in the large intestine, causing mucosal irritation [47] and local inflammation, which lead to intraluminal osmolality changes, and subsequent water influx, resulting in diarrhea [3,26]. This may explain the positive correlation between the greater abundance of *Lactobacillus* and the intestinal disorder. Moreover, *Bifidobacteria* are primary degraders of various complex carbohydrates, such as starch, producing lactic acid, as well [48,49]. However, the prevalence of this taxon was correlated with an increase in heart rate and lameness in horses from the SB treatment group. Dysbiosis can alter gastrointestinal bacterial metabolism and may trigger the release of endotoxins [50,51]. These compounds destroy the intercellular junctions of the intestinal wall [52], leading to an increase in gut permeability and favoring bacterial translocation, which, in turn, may cause systemic disorders [32,53]. Therefore, under some gastrointestinal conditions, high abundance of this taxon might be detrimental [54].

The decrease in fibrolytic bacteria was also more evident in the SB treatment group, and could, at least in part, lead to systemic and intestinal disturbances that can trigger acute laminitis onset. These bacteria are related to the protection of the intestinal epithelium [11,55]. However, it is well known that starch overload causes intestinal dysbiosis, which lead to production of lactic acid by amylolytic bacteria and, consequently, a decrease in pH that can cause the death of fibrolytic bacteria [41,56]. The association between low intestinal pH and the death of these microorganisms may result in severe mucosal damage, compromising the intestinal barrier function [2,21]. Although we did not measure intestinal or fecal pH, it was observed that reduction in relative abundance of these bacteria, mainly from the Lachnospiraceae and Ruminococcacea families, was correlated with the onset of systemic disorders such as diarrhea, hyperthermia, lameness, and inappetence. Previous reports have considered fibrolytic bacteria as markers of digestive eubiosis relevant to the intestinal homeostasis of horses [18,57], since these microorganisms supply most of the host’s energy requirements [17]. Therefore, the absence or reduction of these microorganisms is related to increasing the animals’ vulnerability to conditions such as laminitis [5,58].

We hypothesized that buffer (Mg(OH)_2_ + Al(OH)_3_) solution could have prevented or minimized the microbiota changes and clinical signs of starch overload in horses from the SB treatment group, which did not happen. This buffer has been reported to treat large bowel impaction in horses by oral administration [25,59]. Its effect increases intraluminal osmolality, and subsequent water influx, pushing the impacted compound through the hindgut until it is eliminated [26]. On the other hand, magnesium salts cause irritation of the intestinal mucosa, which can lead to local inflammation and changes of the commensal microbiota [48,60]. Although we did not measure the levels of magnesium ion in our study, we suggested that the buffer, in combination with starch overload, could have intensified and predicted these alterations. It was observed that none of the horses from water-buffer treatment showed any disorders, unlike those from the SB treatment group, that showed microbiota changes. Additionally, clinical signs were observed starting from T12 to T48 in horses from the water–buffer treatment group, while they were noticed just 24 h after the overload in horses from SS treatment, with fecal consistency and smell corroborating findings of a previous publication about starch overload [29] and buffer administration [27]. It is well known that the buffer is considered a cathartic compound [25,61], therefore the intensity of the diarrhea in SB treatment may have resulted from the combined effects of the buffer and the starch overload. Additionally, when the Mg(OH)_2_ + Al(OH)_3_ solution is administered by nasogastric intubation, 20% of the magnesium is absorbed in the small intestine [62,63], which we could not assume in our study, since the buffer was administered directly into the cecum, placing all of the magnesium ion of the solution into the hindgut at first, again suggesting a likely contribution of the buffer to the microbiota changes and clinical signs, mainly diarrhea. Nevertheless, there have been no studies using Mg(OH)_2_ + Al(OH)_3_ intracecally, indicating the need for complementary research involving the measurement of magnesium ions to confirm these hypotheses and to describe all the effects of the activity of magnesium salts in the horses’ intestinal tracts.

Most changes in fecal microbiota lasted 48h following starch overload, similar to the signs of diarrhea, mainly in SS treatment. The constant supply of fiber in the diet (hay) throughout the experiment undoubtedly helped, since dietary fiber increases intestinal and fecal pH by buffering fermentation products in the large intestine [58,64]. In spite of not measuring the fecal pH, we observed that the abundance of amylolytic and fibrolytic bacteria were returning close to the base line (T0). However, the microbiota changes remained stable until 72h after starch overload in SB treatment, probably because of the intensity of the alterations due to the combined effects of buffer and starch overload. In addition, in the first hours of our study, the fiber supply also may have caused a delay in changes in the fecal microbiota, which started 12h after overload, by its buffering activity. Another study found that feces of horses fed with hay showed lower buffering capacity, with consequent pH reduction only after 16h of starch overload [29].

The individual vulnerabilities to intestinal and systemic diseases of horses submitted to different managements (diet, drugs) vary widely [6,17]. We noticed that not all animals from each starch treatment presented clinical alterations of the same intensity. This can be explained due to a difference in gastrointestinal microbiota, as seen clinically in animals, and resulting in individual susceptibility to intestinal diseases [17,65]. Moreover, there are still numerous bacterial species that have not been characterized (44%), as reported in other studies [66,67].

## 5. Conclusions

We conclude that the changes in the fecal microbiota and, consequently, the clinical signs were not stabilized or minimized by the intracecal Mg(OH)_2_ + Al(OH)_3_ solution. On the contrary, the association between this buffer and the corn starch overload intensified microbiota changes and systemic effects, likely due to the cathartic effect of the buffer and the drop in intestinal pH caused by starch overload. These actions lead to deep reduction of some members of the Ruminococcaceae and Lachnospiraceae families, which are essential for gut health, and an increase in amylolytic bacteria, mainly *Lactobacillus*, *Streptococcus*, and *Bifidobacterium* genera. Moreover, dysbiosis was related to gastrointestinal disorders, such as diarrhea, and the onset of systemic effects that can trigger laminitis syndrome, such as hyperthermia, inappetence, high heart rate, and lameness.

## Figures and Tables

**Figure 1 animals-12-03435-f001:**
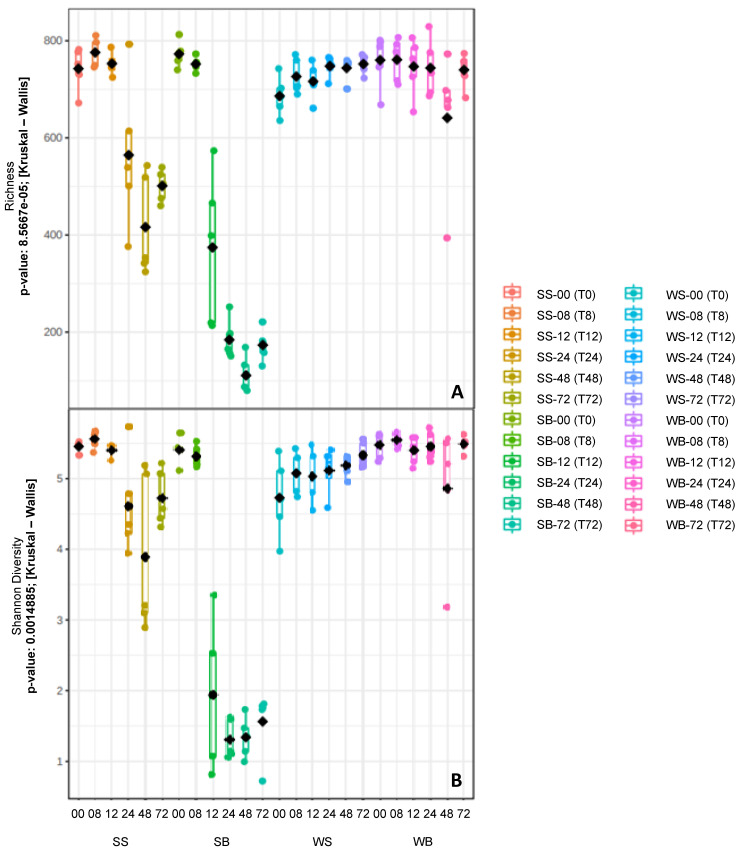
Richness and diversity of bacterial communities predominant in fecal samples 72 h following treatments: water–saline—WS; water–buffer—WB; starch–saline—SS; starch–buffer—SB. T0—before gastric administration of water or starch; T8—08 h; T12—12 h; T24—24 h; T48—48 h; T72—72 h. (**A**) richness (number of species observed); (**B**) diversity by Shannon index.

**Figure 2 animals-12-03435-f002:**
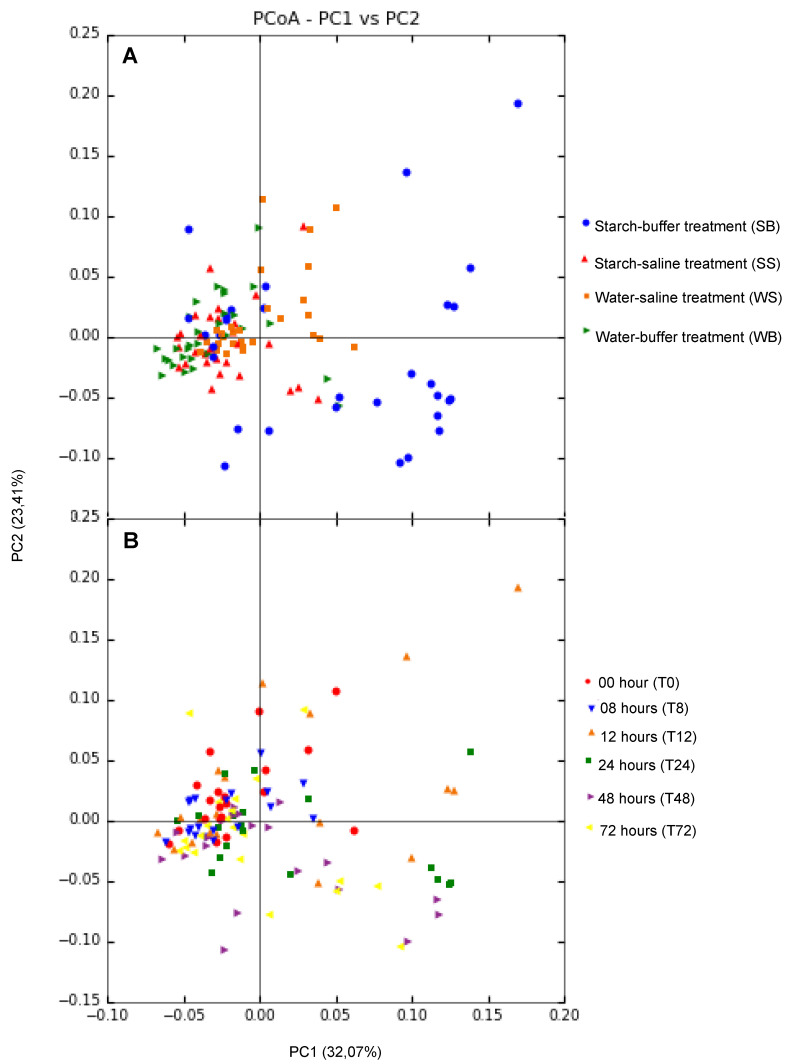
Principal coordinates of the UniFrac weighted distance for the bacterial communities based on treatment and sampling intervals. The UniFrac distance is a mensuration of β-diversity, or the extension of the relation among the bacterial communities. (**A**) treatment; (**B**) sampling interval. The percentages in parentheses in the axis titles represent a variation of 55.48%, explained by the principal coordinates.

**Figure 3 animals-12-03435-f003:**
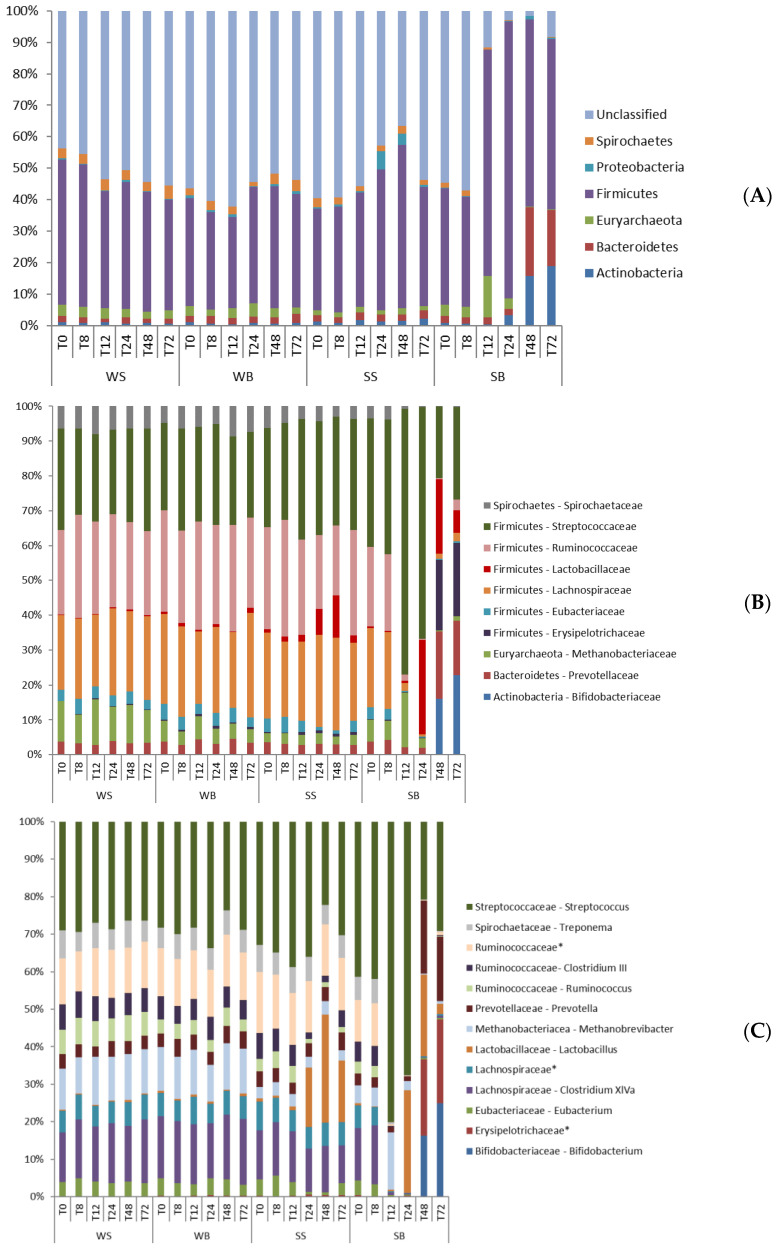
Relative abundance of the predominant bacterial communities (>1%) in the fecal samples 72 h following treatments: water–saline—WS; water–buffer—WB; starch–saline—SS; starch–buffer—SB. T0—before gastric administration of water or starch; T8—8 h; T12—12 h; T24—24 h; T48—48 h; T72—72 h; (**A**)—phyla; (**B**)—families; (**C**)—genera. * Genus not identified belonging to Family mentioned.

**Table 1 animals-12-03435-t001:** Groups and treatments of horses exposed or not to starch overload and/or buffer.

Groups	Treatments	
I (n = 5)	WS	Water (10 L) administered by nasogastric tube and, after 8 hours, 5 L of NaCl 0.9% solution administered intracecally.
	SB	Corn starch overload ^‡^ diluted in water (10 L) administered by nasogastric tube and, after 8 hours, 5 L of a buffer ^†^ diluted in NaCl 0.9% solution administered intracecally.
II (n = 5)	SS	Corn starch overload ^‡^ diluted in water (10 L) administered by nasogastric tube and, after 8 hours, 5 L of NaCl 0.9% solution administered intracecally.
	WB	Water (10 L) administered by nasogastric tube and, after 8 hours, 5 L of a buffer ^†^ diluted in NaCl 0.9% solution administered intracecally.

WS = water–saline; SB = starch–buffer; SS = starch–saline; WB = water–buffer. ^‡^ Maizena Duryea^®^ (17.6 g/kg of body weight; Unilever Brazil Industry Ltd.a., Garanhuns, PE) [29]. ^†^ 3.5 g of aluminum hydroxide (Al(OH)_3_), 65.6 g of magnesium hydroxide (Mg(OH)_2_).

## Data Availability

Publicly available datasets were analyzed in this study. This data can be found here: https://www.ncbi.nlm.nih.gov/bioproject/509648 (accessed on 26 October 2022).

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
