# Peer review of "Effects of Starch Overload and Cecal Buffering on Fecal Microbiota of Horses"

_animals, 2022, doi:10.3390/ani12233435_

Round 1

Author Response

1. The article is clear but I have trouble understanding the experimental scheme. Indeed, you had 10 horses that you divided into two batches which then received two treatments. Why not pass the 10 horses on the 4 treatments with a Latin square? And did you randomize the two treatments within each group?

Response 1. We didn’t choose Latin square to avoid double starch exposition. Horses were randomly allocated into 2 groups (n=5) with 4 blocks of treatment (Water-saline, Water-buffer, Starch-saline, Starch-buffer). There were 15 days of interval among treatments.

2. It is also a pity not to have added groups receiving the buffer or the NaCl orally in order to understand this deleterious effect of the treatment with the buffer by coecal route and to argue your discussion (lines 267-269).

Response 2. Indeed, that is a good purpose. Unfortunately, we had a small group of animals and short budget, so we had to prioritize our hypothesis.

- Minor Points

line 67: is the number 23391/15 that of the ethics committee or that of the study? Response: This study was approved by the Ethics Committee on the Use of Animals (CEUA) of the School of Agricultural and Veterinarian Sciences – FCAV/Unesp-Jaboticabal Campus, under the protocol # 23.391/15.

line 77: during the adaptation period then during the experimental period, what did the horses receive, 2 kg of commercial diet and 8 kg of hay? Only eight kilos of hay? Response: During the 30th day adaptation period, horses had limited access to pasture, and received daily 2 kg of commercial feed (Selvagem® - Agromix – Jaboticabal, SP, Brazil), 4 kg of coast-cross hay (Cynodon dactylon) and water ad libitum. At the beginning of the experimental period, horses were restricted to stables and we standardized a diet with 8 kg of hay per day. 

- Major Points

Why didn't you measure the pH of the feces? Response:

Please present your statistic model: did you analyze the 4 groups together? What do these p< or = at 0.001 correspond to? Please add a table with your statistic model. Response: All groups were evaluated together, and the values above refer to the p value from Kruskal-Wallis statistical analysis using MicrobiomeAnalyst tools (Reference 36).

Why didn't you measure the pH of the feces? Response: We did not measure the fecal pH and we take for granted that the corn starch overload, even in the presence of the buffer, decreased intraluminal pH. We based it on the literature cited which showed a reduction of the intestinal pH after starch overload due to an excessive production of lactic acid by amylolytic bacteria that were increased in our study. Besides, the buffer solution used is characterized as a cathartic product and used to treat ruminal and gastric acidosis according to the literature. However, after the buffer administration intracecal in our study, the microbiota changes and the clinical signs intensified, instead of stabilizing. Therefore, we assumed the fecal pH maintained low after 12 hours in the starch-buffer treatment.

The others considerations by the reviewer are marked up in the manuscript using "Track Changes" function

Reviewer 2 Report

Throughout your discussion, you write many speculations/conjectures that need more theoretical and scientific grounding. Please support your ideas/thoughts with quotes/publications.

Also, I haven't seen any blood tests such as magnesium, hematocrit, and plasma proteins that support your thoughts about dehydration and blood minerals (Mg).

So it would be best if you improve the discussion.

Specific comments. 

Lines 173-176:  What are SC, C, and CB?

Lines: 238 - 240  Please, include the r values for all correlations. 

Line 263:  “induced by magnesium from the buffer (Mg(OH)2 + Al(OH)3)” or by the dose/concentration of the buffer components. 

Lines: 268 - 269:  Did you measure Mg on blood and/or fecal samples? 

Line 270: “total magnesium present” - how much?

Author Response

You will note that the notes to reviewer we have made are inserted in the manuscript and marked up with the "Track Changes" function.

Reviewer 3 Report

The manuscript presents data on the equine microbiome. The animals were challenged by starch and the impact of starch load on the microbiome is elucidated and discussed in the context of clinical observations. The application of a buffer solution is used to look on the effects of starch plus buffer - infusion into the caecum - on the microbial community. For my understanding the inclusion of the buffer application in the study is useles. There is no information about the buffer capacity and the intraintestinal milieu after starch overload and in particular on the buffering effect after application.

Author Response

- Changes were made on the manuscript using "Track Changes" function.

Round 2

Reviewer 2 Report

My last point: Pages 88-89 : please write values o crude protein, fat, fiber and mineral values (magnesium (if have) and phosphorus and calcium.  

Reviewer 3 Report

I see the effort of the authors to revise the manuscript and would accept the manuscript.

However, I still think that the part related to the buffer application is not convincing. The basic question is:

Is a buffer, -entering the caecum - able to limit critical changes in the milieu of the chyme? In this case there was no effect, moreover the condition worsened. Reasons for that can be a insufficient buffer capacity or an acid load that exceeds the buffering capacity.